Review

**Subject Area:**
cellular biology/biochemistry/molecular biology

cell–cell adhesion, cytoskeletal regulators, Rho GTPases, junction assembly, junction integrity, junction recycling

**Author for correspondence:**
Marios Georgiou
e-mail: marios.georgiou@nottingham.ac.uk

# The multifarious regulation of the apical junctional complex

Alexandra D. Rusu and Marios Georgiou

School of Life Sciences, University of Nottingham, Nottingham NG7 2UH, UK

 ADR, 0000-0001-6362-015X; MG, 0000-0003-3321-9124

Epithelial cells form highly organized polarized sheets with characteristic cell morphologies and tissue architecture. Cell–cell adhesion and intercellular communication are prerequisites of such cohesive sheets of cells, and cell connectivity is mediated through several junctional assemblies, namely desmosomes, adherens, tight and gap junctions. These cell–cell junctions form signalling hubs that not only mediate cell–cell adhesion but impact on multiple aspects of cell behaviour, helping to coordinate epithelial cell shape, polarity and function. This review will focus on the tight and adherens junctions, constituents of the apical junctional complex, and aims to provide a comprehensive overview of the complex signalling that underlies junction assembly, integrity and plasticity.

## 1. Introduction

Given the invaluable role cell junctions play in tissue morphogenesis and homeostasis [1–6] it is imperative that epithelial cells form and actively maintain intercellular adherens junctions (AJs) and tight junctions (TJs). In order to cope with the cell shape changes and movements that accompany epithelial development and homeostasis, cell junctions are required to be both strong and plastic. Such plasticity is generated through a constant recycling of AJ and TJ components [7–14].

To ensure homeostasis, cytoskeleton and junction dynamics need to be tightly regulated, both spatially and temporally. This review aims to give a fully comprehensive overview of the composition, function and plasticity of the adherens and TJs, together with a discussion of our current understanding of the regulatory processes and molecular machineries involved in maintaining junction integrity in a living epithelium. The review aims to draw a picture of how multiple protein families work together to modulate cell junction plasticity and the dynamic interactions that occur between cell junctions and the cytoskeleton, highlighting the bidirectionality and looping nature of the regulatory processes. For this purpose, the interplay between AJs, TJs, the actomyosin and microtubule cytoskeletons, Rho GTPases and protein turnover are discussed in detail in relation to their effect on cell junction formation, maturation and maintenance.

## 2. Tight junctions: molecular composition and architecture

TJs encircle epithelial cells at the most apical side of the lateral membrane (figure 1a) where they form a belt-like structure of tight connectivity between adjacent cells, forming the so-called zonula occludens (ZO). TJs act as a paracellular diffusion barrier (or gate) for ions and solutes, and as a fence for proteins that localize at, or in close proximity to, the apicolateral membrane, thus preventing the mixing of apical and basolateral determinants. Additionally, through their linkage to the cytoskeleton they also act as mechanotransducers, forming

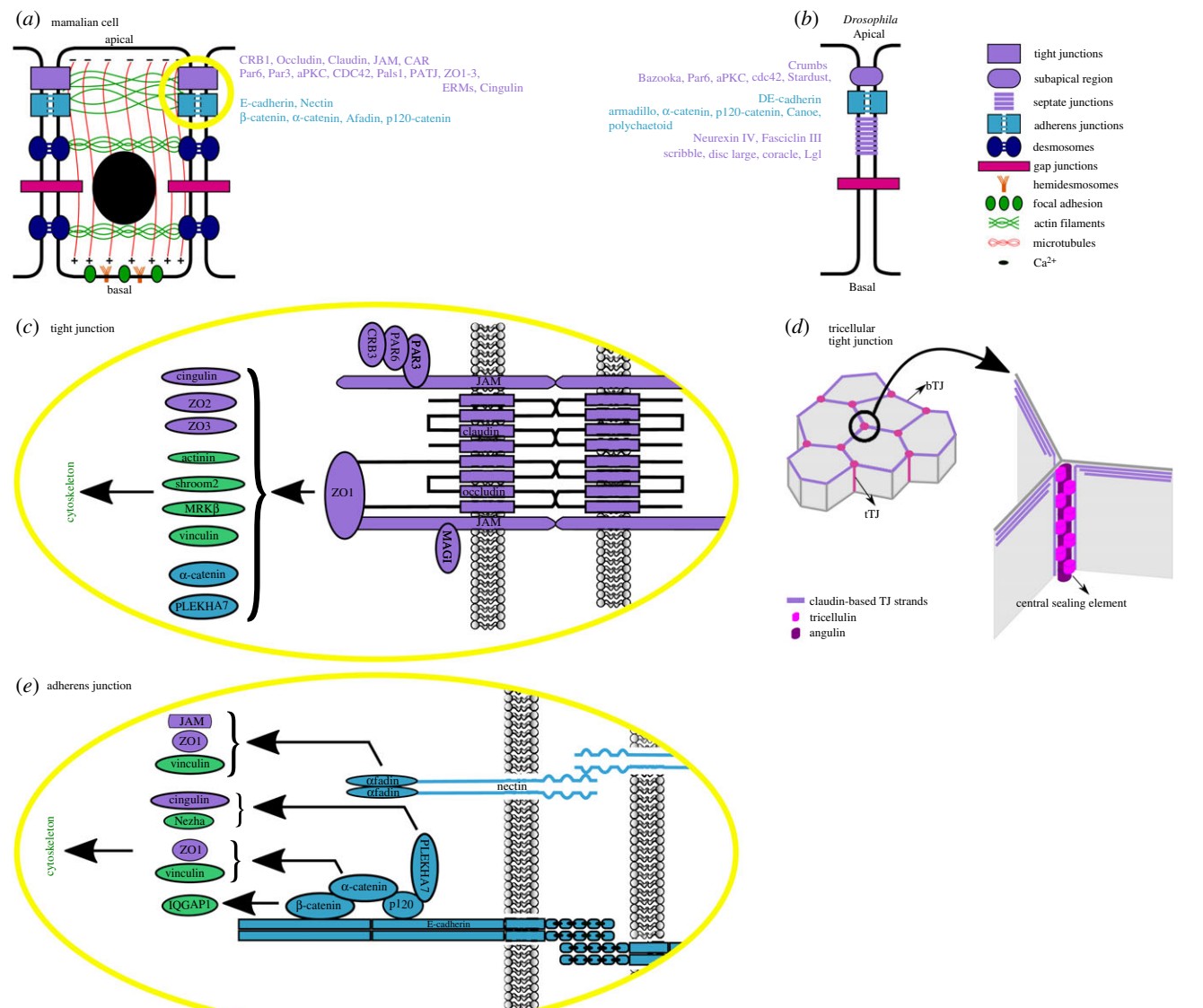

**Figure 1.** Schematic overview of adherens junction, tight junction and cytoskeletal interactions. (*a*) Cell adhesion in mammalian epithelial cells. Communication and adhesion between neighbouring epithelial cells is ensured through the correct apicobasal localization of multiple intercellular junctions. In vertebrates, tight junctions occupy the most apical part of the membrane, followed by adherens junctions, desmosomes and gap junctions. Adhesive contacts between cells and the extracellular matrix are maintained by focal adhesions and hemidesmosomes on the basal membrane. Both cell junctions and focal adhesions are tightly linked to the cytoskeleton through microtubules, actin filaments and microfilaments, forming an intricate and complex network that supports cell signalling. (*b*) Cell adhesion in *Drosophila* epithelial cells. Invertebrate epithelial cells possess adherens junctions that are located apical to the septate junctions, the invertebrate functional orthologue of tight junctions. Homologues of mammalian tight junction proteins can be found in the subapical region of the *Drosophila* cell, a distinct membrane region apical to the ZA. (*c*) Molecular composition and architecture of tight junctions. Claudins, occludins and JAMs are the transmembrane components of tight junctions. They form channel like structures between cells allowing the transport of ions and other molecules. The cytosolic region of TJ transmembrane proteins interact with polarity complexes through Par3, highlighting their role in the establishment apicobasal polarity. A series of scaffolding proteins are also localized at TJs. Their role is to connect TJs to the cytoplasmic plaque which then ensures TJ connection to the cytoskeleton. The most remarkable scaffolding protein is ZO1, which can form complexes with various cytoskeleton linkers and/or modulators such as actinin, Shroom2, MRKβ, vinculin. Equally important, through its interactions with vinculin, afadin and α-catenin, ZO1 confers a link between AJs and TJs. (*d*) Molecular composition and architecture of tricellular tight junctions. Tricellular contacts between cells require a specialized architecture, which is reflected in their molecular composition. tTJ are enriched in tricellulin and angulin in addition to the already presented TJ components. Additionally, TJ strands from neighbouring cells attach and turn basally, forming the central sealing element. (*e*) Molecular composition and architecture of adherens junctions. Type I cadherins, here represented by E-cadherin, are the main component of adherens junctions. E-cadherin molecules can dimerise in both *cis* and *trans* to form cadherin clusters. This homophilic interaction requires $Ca^{2+}$ ions to stiffen the extracellular domain. The cytosolic region contains binding sites for p120- and β-catenin. β-catenin further recruits α-catenin. ZA integrity is maintained through an actomyosin ring, which forms as a consequence of α-catenin's ability to bind F-actin, either directly or indirectly through vinculin. Additionally, p120-catenin further recruits PLEKHA7, which ensures a link to the microtubule cytoskeleton through its interactions with Nezha, a microtubule organizing protein. p120 also binds α-catenin, strengthening the cadherin–catenin complex. Nectin–nectin interactions also take place between neighbouring cells. In the cytosol, nectin forms a complex with afadin which further interacts with vinculin, therefore, providing a link between nectin and the actin network on one side and between nectin and cadherin clusters on the other. Actin and actin interactors are represented in green, AJ components in cyan and tight junction components in purple.

signalling platforms that can alter gene expression [2,4,15,16]. Depolymerization of the cortical actin network [17] or transient ATP depletion [18,19] can disrupt gate function without affecting fence function, thus demonstrating the importance of the cytoskeleton to the TJ and suggesting a mechanistic independence between these two TJ functions.

When viewed using electron microscopy on ultrathin sections, TJs appear as discrete sites of apparent fusion of the outer leaflet of neighbouring plasma membranes [20]. When using freeze-fracture electron microscopy, these sites appear as a network of intramembranous fibrils or strands (commonly known as TJ strands) that interact laterally with strands from adjacent cells [21,22].

TJ biochemical composition includes both transmembrane components (including a large family of claudins, TJ-associated MARVEL domain-containing proteins (TAMPs [23]), and junctional adhesion molecules (JAMs)) and numerous complexes of cytosolic adaptors and regulatory proteins, known as the TJ plaque (figure 1c). The specific composition of transmembrane TJ proteins will affect the barrier properties of the TJ, including its electrical conductance and permeability for ionic and uncharged solutes [24]. The type and concentration of transmembrane TJ proteins will, therefore, differ depending on local transport requirements. The cytoplasmic plaque proteins are involved in several key cellular processes including the regulation of cytoskeletal organization, the establishment of cell polarity and signalling to and from the nucleus [4,25].

Claudins are the major determinant of paracellular permeability [26]. Additionally, claudins have been shown to be sufficient for TJ nucleation, given that when transfected into fibroblasts lacking TJs, claudins are able to induce TJ-like structures [27]. Claudins are able to recruit the ZO scaffolding proteins (ZO-1, -2, -3) and this interaction appears to be required for TJ formation given that ZO deficient mouse epithelial cells are unable to form TJs [28].

In addition to the claudins, occludin, tricellulin and marvelD3 have been identified as constituents of TJ strands [23,29,30]. These proteins have four transmembrane domains and are thought to assemble into heteropolymers, which form the strands. Although occludin was the first TJ transmembrane protein to be identified [29], its specific role at the TJ is still unclear as TJ morphology and barrier function was unaffected in occludin knockout mice [31]. TJ permeability was also unaffected by occludin knockdown in MDCK cells, although knockdown did render cells unable to transduce external and transmembrane signals to the actin cytoskeleton [32].

Within a sheet of cells, the vast majority of junctions that form are between two neighbouring cells (bicellular junctions). However, within epithelial and endothelial sheets, tricellular contacts between cells (the meeting point of three cells) are common and these contacts require several specialized junctional architectures, including tTJs (figure 1d). At tricellular contacts TJ strands from neighbouring cells attach and turn basally, forming the central sealing element [33]. The complex junctional architecture at tTJs is reflected by an altered molecular composition. Both tricellulin and the angulins preferentially localize to tTJs. Although the angulins are a single-pass transmembrane protein family that are not thought to be constituent components of TJ strands, it is thought that the angulins are crucial for tTJ maturation. The angulins localize at primitive tricellular contacts and are required to recruit tricellulin to allow mature tTJs to form [34–36].

The cytosolic TJ plaque comprises scaffold and cytoskeletal proteins. Apart from assisting the membrane proteins in establishing cell–cell adhesion and an ion-selective barrier, this organized network of proteins confers a signalling platform by interlinking TJ components with cytoskeletal proteins and regulators, signalling components (e.g. phosphatases, kinases) and proteins with dual membrane-nuclear localization. Therefore, TJs can regulate vital processes for epithelial architecture such as cell proliferation, invasion and gene expression [37–39]. The best-studied TJ scaffolding proteins are the ZO proteins, which are crucial to establish a link to the cytoskeleton. They play non-redundant functions: ZO1 and ZO2 are embryonic lethal [40,41], while ZO3 appears to be dispensable for TJ formation [42], and depletion of all ZO proteins in epithelial cells render cells unable to form TJs [28]. This is in line with a model which suggests that ZO1 and 2 are the first to be recruited at TJ sites and are responsible for the initiation of TJ formation [28,43]. ZO1 plays a critical role in tissue organization, which is not surprising since ZO1 is a molecular hub, being able to interact with ZO2, ZO3, occludin, claudin, α-actinin, kinases, phosphatases and transcription factors [28,39,41,44–48]. Slightly further away from the membrane, the adaptor protein cingulin links JAMs, occludin, ZO1 and the cytoskeleton [1,49–56]. It plays both structural and signalling functions and has been implicated in cell migration and gene expression regulation [1,57–59]. However, in line with many knockout and knockdown experiments where single adaptor proteins have been targeted, cingulin has been shown to be dispensable for both TJ formation and actin cytoskeleton architecture [57,60]. It is likely that the large number of adaptor and signalling proteins that make up the TJ plaque incorporate extensive redundancy. Other scaffold proteins that localize at least in part to TJs and contribute to the TJ signalling platform are the membrane-associated guanylate kinase inverted (MAGI) proteins, PALS1-associated tight junction (PATJ), paracingulin and multi-PDZ domain protein (MPDZ) [53,61].

Although TJs are vertebrate specific, septate junctions (SJs) function as the main occluding junctions in invertebrates. First identified in *Drosophila* salivary glands [62], SJs are thought to be analogous to TJs from a functional point of view, but they do differ in numerous ways, including in both their molecular composition and in their location on the lateral membrane (figure 1b) [4].

## 3. Adherens junctions: molecular composition and architecture

Although a highly characteristic feature of epithelial sheets, the AJ is not limited to this tissue and can additionally be found in non-epithelial cells. In epithelial cells, AJs are strong anchoring junctions that form the zonula adherens (ZA), just basal to the ZO in vertebrates. Together with TJs and desmosomes, AJs define the apical junctional complex [20]. Just as with TJs, AJs are composed of membrane-spanning adhesion molecules whose extracellular domains form intercellular connections with neighbouring cells, and whose intracellular domains associate with various cytoplasmic scaffolding and signalling complexes, forming a cytoplasmic signalling plaque [63]. Cadherin and nectin-based interactions form the basis of the intercellular connections between cells [64,65] (figure 1e). The backbone of the AJ consists of the cadherin–catenin complex (E-cadherin, β-catenin, α-catenin, p120-catenin) which acts in concert with the nectin–afadin complex to create a tightly regulated

royalsocietypublishing.org/journal/rsob Open Biol. **10**: 190278

signalling platform that internalizes stimuli from outside the cell and returns responses by physically linking the junction to the cell's cytoskeleton [64,66–68]. ZO1 (of the TJ) and α-catenin (of the AJ) are key mediators in linking the apical junctional complex to the cell's actomyosin cytoskeleton by interacting both directly and indirectly with F-actin [47,48,69–73] (figure 1). The actomyosin network forms a belt-like ring around the cell perimeter at the ZA that provides junctional tension, which is required to maintain junction stability [8]. Similarly, PLEKHA7 and p120-catenin (p120ctn) also play a key role in connecting the apical cell junctions to the microtubule cytoskeleton [74]. These AJ-TJ-cytoskeleton interactions are crucial for transducing chemical and mechanical signals from the local microenvironment. These interactions are also responsible for the rearrangement of cell junctions in response to both internal and external stimuli, which is important to maintain epithelial integrity in the face of the constant changes in cell packing that accompany changes in tissue organization, cell division, cell death and delamination [6].

AJs are extremely important for tissue morphogenesis, homeostasis and plasticity. They not only mediate cell adhesion but also cytoskeletal dynamics, intracellular signalling and transcription regulation (mainly through β-catenin). This places them at the core of cellular processes such as cell recognition, cell division, polarization and motility [6,75–79].

From a structural point of view, the AJ consists of a mosaic of independent cadherin and nectin clusters, linked together and coordinated by actin filaments [64] (figure 1e). The cadherin clusters are the major components of the AJ and consist of cadherin–catenin multi-protein complexes. Type I classic cadherins, such as E-cadherin (characteristic of epithelial tissue), VE-cadherin (endothelial tissue) and N-cadherin (neural tissue), are at the core of these complexes and for convenience will be simply referred to as cadherins from now on. They are transmembrane proteins that consist of 5 extracellular domains capable of binding $Ca^{2+}$: a transmembrane domain and a cytoplasmic tail containing a juxtamembrane domain (JMD) alongside a catenin-binding domain (CBD) [80–82]. The extracellular domains of cadherins from adjoining cells form dimers through $Ca^{2+}$-dependent homophilic interactions and therefore physically connect neighbouring cells [83]. However, to maintain a stable cell–cell contact, E-cadherin needs to bind to catenins using its intracellular domain [84]. The JMD contains the binding site for p120ctn, an interaction required for the stabilization of the complex at the cell surface (p120ctn linkage to E-cadherin impairs the association of ubiquitin ligases and endocytic machinery to E-cadherin) [85]. In addition, p120ctn regulates the RhoA and Rac1 RhoGTPases [86,87] and interacts with the transcription repressor Kaiso, a modulator of non-canonical Wnt signalling [88]. The CBD mediates the interaction between E-cadherin and β-catenin. This interaction is important for three reasons: (1) it recruits α-catenin to the AJ; (2) it aids in the transduction of signals from the cell–cell junction to the nucleus [77–79,89]; (3) it modulates canonical Wnt signalling by preventing β-catenin interaction with Tcf/LEF-1 transcription factors [90,91]. It is likely that β-catenin's key role in maintaining cell–cell adhesion is to act as a bridge between E-cadherin and α-catenin molecules, since a DE-cadherin/α-catenin fusion protein is fully functional, even in the absence of β-catenin in *Drosophila* oocytes [92].

Recruitment of α-catenin to the AJ is indispensable for strong adhesion and signal transduction, largely because α-catenin plays a dual role at the junction: it strengthens the E-cadherin–p120ctn interaction by binding p120ctn [93] while also conferring a link between the AJ and the actin cytoskeleton. Less well-studied members of the cadherin–catenin junctional complex include vinculin and EPLIN. Both α- and β-catenin are able to recruit vinculin to the AJ, and vinculin recruitment appears to be crucial for the maintenance of E-cadherin levels at the membrane and for mechanotransduction [66–68]. Taguchi *et al.* also highlight EPLIN's involvement in this process [94]. However, EPLIN's role is likely to be secondary to other mechanisms, given that not all cadherin–catenin complexes link to EPLIN [72].

Cadherin–catenin complexes group in adhesive clusters through a chain of cooperative *cis*- and *trans*-interactions between cadherin ectodomains. Multiple cycles of cadherin dimer assembly and disassembly have to take place in order for the cadherin clusters to form a mature AJ [95–97]. Additionally, in a mature junction, the cadherin–catenin complex interacts with nectins and adhesion molecules from the TJ [64].

Nectins are ubiquitously expressed $Ca^{2+}$-independent immunoglobulin-like transmembrane adhesive receptors. The nectin family comprises of four members, nectin-1 to -4, as well as several splice variants. All members (except nectin-1γ, which is a secreted protein) comprise three Ig-like domains in the extracellular region, a transmembrane region and a cytoplasmic region [98]. Like E-cadherin, nectins can form both *cis*- and *trans*-dimers via the extracellular region, with *cis*-dimer formation being a prerequisite for *trans*-dimer formation between cells [99]. Although each member of the nectin family can form homo-trans-dimers, certain family members can also form hetero-trans-dimers (e.g. nectin-3 with nectin-1 or -2) [99]. The cytoplasmic region mediates nectin's link to cadherin complexes and the cytoskeleton by binding afadin, which in turn interacts directly and indirectly (through vinculin) with α-catenin and the actin cytoskeleton [65]. By interacting with integrins, the nectin–afadin complex also instigates a link between cell–cell and cell–matrix adhesion systems [65].

The importance of the nectin–afadin complex for AJ formation and stability is still under debate. While some researchers suggest that nectins are required for cadherin nucleation [65,100], other observations suggest a dispensable role for nectins and afadin in AJ formation. For instance, afadin deficient mouse embryos show developmental defects and impaired AJ and TJ organization in the ectoderm, however, they are able to establish all three germ layers [101,102]. By contrast, the E-cadherin knockout is lethal in mouse embryos, which fail to form the trophectoderm [103,104]. More recently, Indra *et al.* showed that while all AJs contain cadherin–catenin clusters, only a subpopulation additionally express nectin [64]. Even though the nectin–afadin clusters might be dispensable for AJ formation, they are able to affect the kinetics and dynamics of AJ formation and they can change the adhesive properties of the junctions [64,105]. Fortugno *et al.* show that altered nectin-4 function in human keratinocytes does not impair AJ formation, but delays E-cadherin recruitment at the cell–cell interface affecting junction stability [105]. Nectin clusters are also involved in cytoskeleton regulation and Rho GTPase recruitment, characteristics that will be discussed later.

royalsocietypublishing.org/journal/rsob Open Biol. 10: 190278

PLEKHA7 is a more recently characterized component of the AJs. Localized to the cytoplasmic plaque, PLEKHA7 links the AJ to the microtubule cytoskeleton. Often overlooked because it is dispensable for an organism's viability, PLEKHA7 is important for epithelial tissue homeostasis as it is involved in junction formation, organization, stabilization (at both cadherin and nectin interfaces) and signal transduction [106,107].

Although the cadherin–catenin complex has been extensively studied for the past three decades, many AJ components have been relatively overlooked. A more comprehensive understanding of all AJ components will allow for a better interpretation of how differences in AJ composition can influence AJ adhesion, stability and dynamics, as well as wider effects on cell behaviour, and responses to diverse stimuli.

## 4. The molecular nature of adherens and tight junction interactions—interplay at cell–cell junction assembly

There is extensive interplay between components of the adherens and TJs. For example, ZO1 can interact independently with cortactin, VASP and vinculin, all of which are actin linkers that reside at the AJ [108–110]. ZO1 can also interact with both afadin and α-catenin [111–114]. The ZO1–α-catenin interaction might also explain the ability of ZO1 to modulate β-catenin transcriptional activity [69]. Further examples of AJ-TJ component interactions include: α-catenin binding to ZO2 and ZO3; afadin interactions with JAMs; PLEKHA7 forming a complex with ZO1 and cingulin [74,115–117].

Cell junction formation (figure 2) is a complex multistep process tightly controlled by several signalling pathways that hinge on extensive reorganization of the actin cytoskeleton and ultimately leads to the polarization of the cell. The mechanism employed is mostly conserved, showing few variations between different epithelial cell types and even organisms [118]. The process begins with AJ establishment, driven by initial contacts between two neighbouring cells, followed by contact expansion, which is promoted through Rac1 activity [6]. Concomitantly TJs begin to form. Later, cell polarization is triggered and a switch from Rac1 to RhoA signalling then drives junction maturation. This multistep process can be summarized as follows:

(1) The Rho family GTPase Rac promotes the formation of de novo cell–cell contacts between neighbouring cells by inducing local membrane remodelling, which promotes the formation of both lamellipodia and filopodia. This brings two opposing cell membranes into close proximity [6].
(2) Nectins form homophilic interactions between adjacent cells, which drive the recruitment of already formed cadherin–catenin complexes in an afadin-dependent manner (afadin interacts directly with α-catenin and indirectly with p120ctn) [65,119–122] (figure 2a).
(3) The first cell–cell adhesive contacts are consolidated by the trans-dimerization of small E-cadherin clusters on adjacent cells, which are linked to bundles of actin filaments. The clusters will then expand to form highly dynamic spot AJs [123–125] (figure 2b). Super-resolution microscopy has shown that E-cadherin clusters are formed in a cell–cell contact-independent manner and that they serve as the basic unit for cell–cell adhesion [126].
(4) The establishment of the nascent spot AJs leads to the formation of TJs, with ZO1 being central to this process (figure 2b). Importantly, ZO1 associates with AJs during assembly but later localizes exclusively to TJs [112]. E-cadherin clustering promotes ZO1 localization at sites of cell–cell contact [112,127]. ZO1 links TJ formation to both cadherin and nectin adhesion initiation sites: ZO1 interacts with α-catenin, which forms a complex that is recruited to the primordial junction; ZO1 is also recruited to nectin rich areas through ZO1–afadin interactions where it further recruits JAM proteins [112,128–130]. Through experiments that reconstruct AJs and TJs in fibroblasts, the Takai group showed that JAM recruitment at junction sites takes place basally to nectin, and that the proper TJ-AJ configuration and polarization is triggered by Crumbs recruitment at nectin–cadherin clusters [120].
(5) Subsequently, further E-cadherin clusters form and densely pack together. Without fully connecting to each other, these clusters encircle the cell to form the ZA [126]. Junction maturation (figure 2c) and junction separation processes rely on a complex signalling network involving kinases, phosphatases and GTPase triggers, and are characterized by a switch from Rac to Rho signalling [2,125,131,132].
(6) The junction formation process culminates with the establishment of an apical actomyosin ring linked to the junctions which further supports their dynamic nature, where a constant disassembly and reassembly of AJs and TJs is required [118]. We direct the reader to a number of excellent articles for a more detailed description of junction assembly [123–125,133].

In summary, the adherens and TJs can no longer be considered as discrete complexes, as once thought. AJ and TJ components can interact with each other and modulate each other's activity. Further interdependency is evident when considering interactions with actin-binding proteins, polarity proteins and signalling molecules, all of which are important to regulate junction assembly, maturation, dynamics and positioning.

## 5. Molecular nature of cell junction-cytoskeleton interactions

Both AJs and TJs are tightly linked to the cytoskeleton through actin and microtubule filaments (figure 1). The resulting network links the intercellular junctions to virtually all compartments of the cell, forming a complex signalling platform able to transduce both adhesive and mechanical signals from the membrane into the cell, and back again, as these signals can drive junction reorganization. For example, E-cadherin is required for ECT2 junctional localization and therefore RhoA signalling. In turn, RhoA signalling regulates E-cadherin mobility, by acting on Myosin-II (Myo-II) [134–136]. Furthermore, cadherin cluster assembly and stability, and consequently AJ stability, is greatly increased upon anchorage to the actin cytoskeleton [137], highlighting the interdependence between these two dynamic structures. Cortical actomyosin is particularly important for AJ formation [138–140]; however, AJs are also physically linked to

royalsocietypublishing.org/journal/rsob    Open Biol. 10: 190278

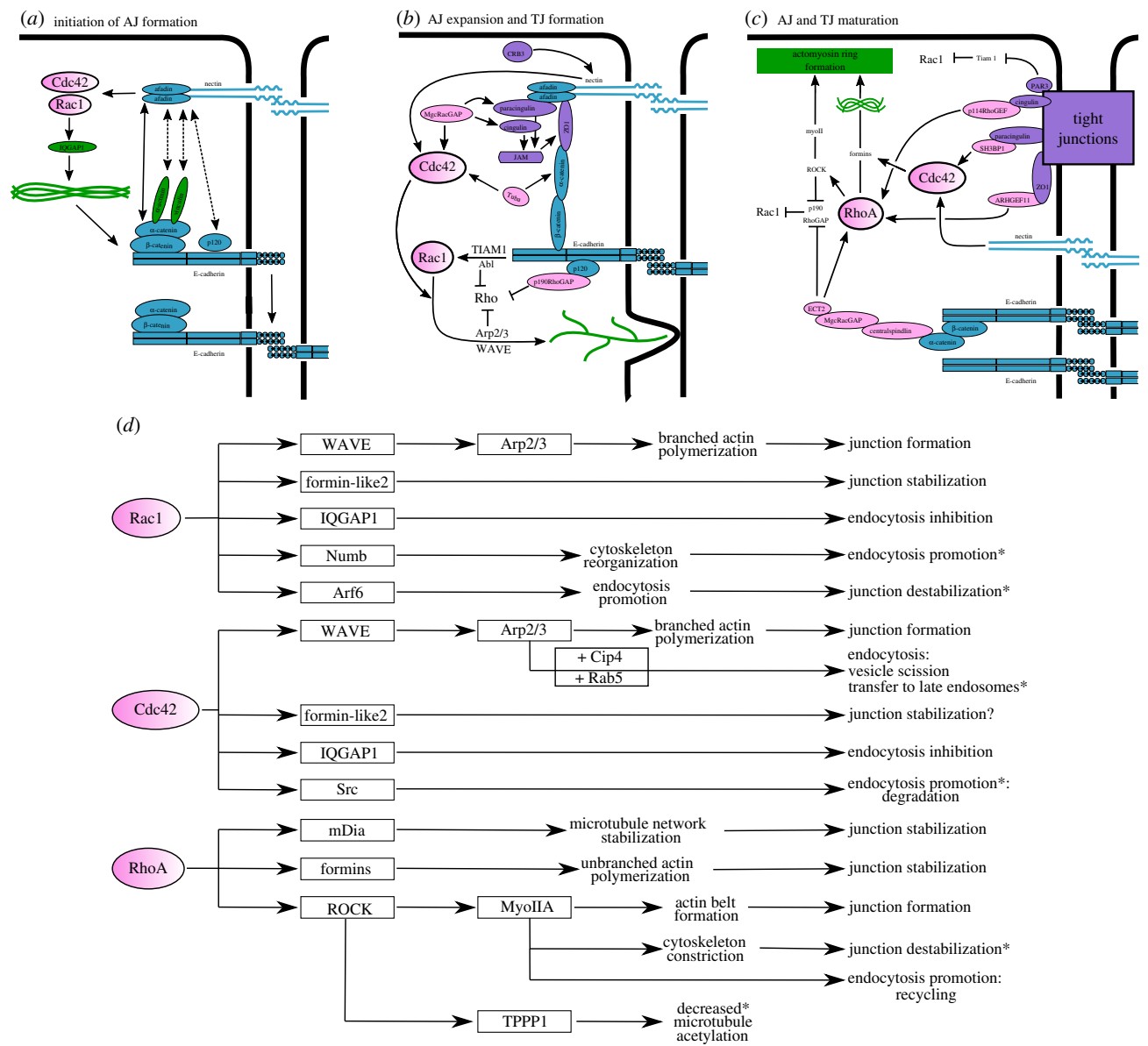

**Figure 2.** Model of cell junction formation and maturation under RhoGTPase control in epithelial cells. (*a*) Initiation of AJ formation. Nectin is recruited at cell–cell adhesion sites. The interaction between nectin molecules drives Rac1 and Cdc42 activation, which in turn modulate IQGAP1. Further, nectin interacts with afadin, which indirectly interacts with vinculin. Consequently, cytoskeleton reorganization occurs driving recruitment of already formed cadherin–catenin complexes to the cell adhesion sites. The recruitment of the cadherin–catenin complexes is aided by afadin's direct interaction with α-catenin and indirect interaction with vinculin, α-actinin and p120-catenin, which increases cadherin adhesion by driving further interactions between cadherin extracellular domains on adjacent cells. (*b*) Adherens junction expansion and initiation of tight junction formation. Cadherin–cadherin interactions activate Rac1 signalling by recruiting Abl kinase and the RacGEF, TIAM1. Nectin interactions at the cell–cell interface activate Cdc42. In turn, Rac1 and Cdc42 activate WAVE and Arp2/3, promoting branched actin polymerization and lamellipodial formation, thereby promoting further cell–cell contact and cadherin clustering. Rac1 also promotes p190RhoGAP recruitment to cadherin clusters, which together with Abl and Arp2/3, ensure Rho signal silencing. While cadherin–catenin clusters start to form, ZO1 is recruited to the primordial junction through α-catenin and/or afadin. ZO1 then triggers TJ formation by recruiting JAMs. Tuba and MgcRacGAP are also recruited to junction formation sites and further promote Cdc42 activation. Later, nectin–cadherin clusters recruit Crumbs (CRB3) which triggers the rearrangement of protein complexes and cell polarization. (*c*) Adherens and tight junction maturation requires a switch from Rac1 to RhoA signalling. Par3 is recruited to nascent TJ sites where it inhibits TIAM1 and therefore Rac1 signalling. Additionally, α-catenin recruits the centralspindlin complex, which simultaneously activates RhoA and inactivates Rac1 by inhibiting p190RhoGAP localization at AJs. RhoA is also activated at TJs by p114RhoGEF and ARHGEF11 recruitment. Once activated, RhoA acts on formins and Myo-IIA (through moderate ROCK activation) promoting the formation of an actomyosin ring at the ZA. (*d*) Summary of pathways acting downstream of RhoGTPases Rac1, Cdc42 and RhoA, which regulate cell adhesion by modulating adherens and tight junction formation, maturation and stability. All three proteins regulate junction assembly and stability by modulating cytoskeleton dynamics and protein turnover. Depending on the circumstances (cell type, process regulated, timing, pathway on which they act) any of them can promote junction assembly and stabilization or junction disassembly. Asterisk marks processes that lead to loss of cell adhesion and increased cell spreading, migration, or invasion.

actomyosin networks found in the medial–apical cell cortex [141,142], allowing actomyosin-generated forces to modulate cadherin cluster distribution [143] and consequently AJ adhesion properties.

## 5.1. AJs, TJs and the actomyosin cytoskeleton

AJ stability relies on the actomyosin cytoskeleton. Several studies have shown that actin or myosin depletion can

disrupt cadherin-based cell–cell adhesion, suggesting a complex interplay between the AJ and cytoskeletal components [138–140,144–147]. Cadherin–catenin complexes, Nectin–afadin complexes, TJs and the actin cytoskeleton all collaborate to organize apical intercellular junctions (figure 2), but the cadherin–catenin complex alone, in association with vinculin, is sufficient to induce cell–cell adhesion under actomyosin contractile forces [148]. The link to actomyosin strengthens adhesion by stabilizing E-cadherin and promoting cadherin clustering [134]. One molecular component of the cytoskeleton essential for junction integrity is Myo-II, which has 2 isoforms present at cell–cell junctions in mammalian epithelial cells. It has been shown that depletion of either Myo-IIA or Myo-IIB can perturb E-cadherin distribution at junctions, with Myo-IIA depletion being able to additionally impair E-cadherin recruitment at AJs [140]. However, both motor proteins had to be depleted simultaneously in order to disrupt ZO1 distribution at the ZO. Myo-IIA proved essential for cell–cell homophilic adhesion, while Myo-IIB ensured junction integrity by modulating apicolateral tension and the dynamics of actin filaments [140]. Actomyosin pulses driven by Myo-II contraction can also deform cell junctions by modulating E-cadherin distribution, suggesting that a physical link between E-cadherin and actomyosin must exist. It is clear that the actomyosin component linked to AJs is F-actin; however, it is still under debate how this link exactly occurs. Multiple lines of evidence show that α-catenin plays an important role: it has been shown to link actin filaments both directly and indirectly through vinculin and EPLIN [70–72]. Vinculin has been shown to have a non-uniform distribution at junctions in the *Drosophila* embryonic ectoderm, where it has a planar polarized distribution, mirroring the distribution of Myo-II [149].

It has been 24 years since α-catenin was first proposed to link AJs to the cytoskeleton; however, the molecular mechanism behind this link is arguably still not fully elucidated. In 1995, Rimm *et al.* showed that α-catenin can bind actin filaments *in vitro* and therefore proposed a model for AJs in which β-catenin links E-cadherin to α-catenin, which then provides a link to the actin cytoskeleton [150]. This model became dogma until 2005 when the Weiss and Nelson groups showed that, at least *in vitro*, α-catenin cannot bind actin filaments and β-catenin simultaneously, not even indirectly via vinculin or α-actinin [151,152]. However, even though a cadherin-β-catenin–α-catenin–F-actin complex has never been immunoprecipitated nor assembled *in vitro* so far, recent research promotes the idea of α-catenin providing a link between cadherin–catenin complexes and the actin cytoskeleton. For instance, a study performed in *Caenorhabditis elegans* highlights the need for such a complex *in vivo* showing that truncations of α-catenin or F-actin, which would impede their ability to bind, disrupt *C. elegans* development [153]. A further study performed in *Drosophila* also supports the view that the cadherin–catenin complex has to be linked to the cytoskeleton by monomeric α-catenin [154]. Yonemura *et al.* show that α-catenin conformation can be modified by force, and that the conformation that α-catenin adopts under tension enables its interaction with vinculin [70], a finding reinforced by studies performed in the De Rooij lab [148]. In 2013, a model in which α-catenin can switch from a 'closed' to an 'open' conformation that allows for vinculin binding was finally described by Isiyama *et al.* Their model relies on α-catenin being able to simultaneously bind β-catenin and F-actin as the tension needed to switch to an 'open' conformation is driven by actomyosin contraction. In this model, vinculin reinforces the cadherin–catenin–actomyosin connection, rather than establishes it [155]. Since then, further studies that performed α-catenin stretching experiments, using magnetic tweezers, or combined magnetic twisting cytometry with a FRET-based α-catenin conformation sensor, have reinforced this model [156,157]. Moreover, Buckley *et al.* used a single-molecule optical trap-based assay that replicates the architecture and forces present at AJs to show that a cadherin–catenin complex consisting of only E-cadherin, β- and α-catenin robustly links to the actin cytoskeleton [158]. Recently, it was shown that α-catenin directly binds actin through its actin-binding domain (αABD). Using point mutants in this domain it was shown that the αABD–actin interaction stabilizes AJs and initiates vinculin recruitment [159]. It was further shown that AJ dynamics are affected by whether the AJ–cytoskeleton link is through a direct αABD–actin interaction, or through vinculin, with the former direct interaction resulting in increased AJ dynamics [159].

It is, however, important to note the many other ways in which AJs can be linked to the actin cytoskeleton. For example, E-cadherin directly interacts with the motor Myosin VI, which is involved in processes such as Clathrin-mediated endocytosis and cell migration [160,161]. Anillin is a scaffolding protein that links cell junctions to the actin cytoskeleton, and which has recently been shown to be able to regulate E-cadherin and β-catenin expression in *Drosophila* [162]. Although it is not required for junction formation, a lack of anillin leads to severe disarrangement of the actomyosin belt at the ZA, ultimately leading to AJ and TJ disassembly, as shown by transient knockdown in human prostate, colonic, and lung epithelial cells [163]. Tricellulin has also been shown to recruit both F-actin and Myo-IIB to cell vertices [164]. Afadin and EPLIN are other ZA components that can mediate the AJ–F-actin link [72]. Although by no means fully understood, one theory is that the multiple interactions between AJ components and cortical F-actin could be modulating the adhesive, dynamic and mechanical properties of the AJ [134].

For TJs, the most important cytoskeletal linkers are the ZO proteins. Fanning *et al.* showed in 1998 that ZO1 links directly with F-actin and also with ZO2 and occludin [47], which can also bind directly to F-actin [48,73]. ZO1 also interacts with Shroom2 and MRCKβ, both regulators of Myo-II activity [165,166]. Additionally, cingulin, another TJ component, was shown to interact with both myosin and F-actin [55,56]. Therefore, just as with AJ–F-actin interactions, there exist multiple TJ–cytoskeleton interactions, which may, in turn, provide TJs with the versatility needed for their barrier function [167].

ZO1 knockdown in epithelial cells causes changes in actin and myosin organization at the ZO [168,169]. According to Spadaro *et al.* [170], when ZO1/ZO2 heterodimers localize at junctions, ZO1 is kept in an active stretched conformation through interactions with both junctional components and actomyosin. This displays their ZPSG domain and as a consequence facilitates their interaction with the transcription factor DbpA, and occludin, which in turn, leads to the control of cell proliferation rate and a strengthening of the TJ barrier function, respectively [170]. This provides a further example of how actomyosin contractile forces at the junction are required for correct junction formation and function.

The actin cytoskeleton is a highly dynamic structure [151] and therefore filament turnover and actin assembly must take place at the ZA in order to assure cytoskeleton integrity. Indeed, several actin regulators, such as Arp2/3, cortactin and formins, are also recruited to the AJs, which confer cadherin clusters the ability to actively build actin filaments [134]. Interestingly, α-catenin was reported to supress Arp2/3 and cofilin activity by either substrate competition or by inducing changes in actin filament conformation [152,171]. Another protein involved in this post-nucleation reorganization of the actin cytoskeleton is N-WASP. Other than regulating Arp2/3 activity, N-WASP is required for actin bundle stabilization at the ZA and therefore for maintaining ZA integrity [172,173]. In turn, both Arp2/3 and N-WASP are required for AJ and TJ assembly and proper organization: N-WASP inhibition leads to disruptions to the E-cadherin apical ring and can perturb AJ and TJ formation [172,174].

## 5.2. AJs, TJs and the microtubule cytoskeleton

Interphase microtubule (MT) organization can range from radial patterns, with MT plus-ends projecting towards the periphery, to parallel 'alligned MTs', which run parallel to the lateral junctions [175–178]. Interactions between MTs and the cell cortex are key to positioning the microtubule array and in generating and sustaining cell polarity in polarized cells [179]. Epithelial cells contain stable microtubules that are not associated with centrosomes, which are required for epithelial cell apicobasal polarization. Apicobasal polarization, the generation of distinct apical and basolateral cortical domains, is accompanied by the development of a non-centrosomal apicobasal MT array, with MT minus ends located apically. It is thought that minus end-anchoring proteins (e.g. γ-tubulin, ninein and Nezha) and plus end-tracking proteins (+TIPs; e.g. the CLASP-LL5 complex) are recruited to the apical and basal regions respectively, and are required to correctly position and align microtubules along the apicobasal axis [180].

AJ homeostasis also depends on MT integrity, since MT depolymerization or altered plus-end dynamics disrupt AJ architecture [177,178]. PLEKHA7 links AJs to microtubules by simultaneously binding p120ctn and Nezha (a MT minus end binding protein) [178]. Nezha further interacts with KIFC3 to promote ZA formation [178].

MTs and MT motor proteins seem to be important in the assembly of the AJ. The projection of microtubules to epithelial AJs is enhanced during junction assembly [176]. The interaction between kinesin, a plus-end-directed microtubule motor, and p120ctn has also been shown to be critical for junction assembly [181,182]. Dynein, an MT motor protein that can bind β-catenin, was demonstrated to be responsible for tethering MTs at the junction [176,183]. It was further shown, using anti-dynein antibodies, that dynein disruption inhibits junction formation [176]. It has been suggested that MTs, tethered to the AJ via dynein, provide a track for the kinesin-dependent delivery of the junctional components necessary for junction formation [176].

Other linkers that could potentially tether MTs to the AJ include IQGAP1, which interacts with several +TIPs, and mDia, a formin that regulates actin organization at the AJ and can also stabilize MTs through an actin-independent mechanism. It is, however, highly likely that microtubule and actin dynamics at the ZA are interlinked [167,176,183,184].

Research in the past two decades has not focussed extensively on the microtubule–TJ relationship, despite evidence back in 1995 showing that a prolonged perturbation of MTs can induce a loss of TJ barrier function [185]. A more recent study showed that the loss of barrier function that follows TJ physical disruption, provoked by ZO1 relocalization, is in part microtubule-dependent [186]. Additionally, MTs were shown to modulate TJ component turnover [186,187].

## 6. The importance of Rho family GTPases for cell junction formation and maturation

As described above, the formation and maintenance of cell–cell junctions requires an intimate relationship between AJ and TJ protein complexes and the cell's cytoskeletal components and regulators. However, the Rho family GTPases are another protein family key to this process. There is extensive evidence linking the activity of the canonical Rho GTPases, Rho, Rac and Cdc42 with cell junction regulation [6,63,188] (figure 2d). The intricate interplay between these protein families is bidirectional, meaning that not only do the Rho GTPases regulate junction formation, maturation, maintenance and dynamics, but also junction components balance, in turn, the activity of the Rho GTPases. This regulation of Rho GTPase activity therefore provides a mechanism whereby intercellular junctions can influence cell morphology and cell polarity. Generally, this regulation is achieved via the activation/inactivation of rho/rac/cdc42 specific guanine nucleotide exchange factors (GEFs) and GTPase-activating proteins (GAPs) [189], as discussed in the following paragraphs.

The main Rho GTPase that modulates junction formation is Rac (figure 2), which is activated by Abl kinase and the Rac-specific GEF TIAM1. TIAM1 is recruited to cadherin–cadherin interaction sites and can, in turn, activate Arp2/3 by recruiting its activator, WAVE and Formin-like 2. This leads to branched actin polymerization favouring lamellipodial formation and further cell–cell contact establishment within and surrounding pre-existing cadherin–catenin clusters [190–193]. Simultaneously, Rac will facilitate p190RhoGAP recruitment to the cadherin adhesion sites via p120ctn, which together with Abl kinase assures Rho signal silencing [190,194]. Meanwhile, cdc42 is activated by nectin–nectin interactions at cell contacts and/or by ZO-1 and tuba interactions. This leads to N-WASP and Arp2/3 activation and a consequent enhancement of actin polymerization [164,195,196]. An alternative mechanism to ensure simultaneous Rac activation and Rho inhibition is provided by paracingulin, a component of both adherens and TJs in some cell types, which can both activate TIAM1 and inactivate ARHGEF2 [197].

As the AJ matures, there is a switch from Rac to Rho signalling (figure 2b). The cadherin–catenin complex recruits Par3 which leads to TIAM1, and therefore Rac, inactivation allowing for TJ formation and AJ maturation [198]. Par3 retention at the ZA can also be promoted by cdc42 and Pak4 activity [199,200]. In addition, α-catenin has been shown to localize centralspindlin, which subsequently recruits the RhoGEF ECT2 to the junctional cortex, thereby promoting Rho signalling at the ZA. Centralspindlin also recruits MgcRacGAP and inhibits the junctional recruitment of p190B RhoGAP. This activation of Rho and inhibition of

Rac signalling has been shown to generate junctional tension, which supports junction integrity, via myosin IIA [201]. Additionally, Myo-IIA interacts with Rho-associated protein kinase (ROCK) at the ZA, which promotes RhoA activity at the junction by preventing the cortical recruitment of p190B RhoGAP [136,202].

TJ maturation is driven in a similar fashion (figure 2b): cingulin recruits p114 RhoGEF which activates Rho signalling and leads to Myo-IIA recruitment to the TJs [203]. Similarly ZO1 can recruit ARHGEF11 to TJs which activates Myo-IIA in a Rho-dependent manner facilitating cell junction formation [204]. This suggests that distinct GEFs, localizing to the AJ and TJ, may activate different pools of Rho with the effect of supporting the stability of the apical junctional complex as a whole.

Anillin is a scaffolding protein that stabilizes actomyosin cables and was previously thought to function primarily during cytokinesis [205,206]. A recent paper, however, has found that a population of Anillin localizes to cell–cell junctions throughout the cell cycle [207]. Using *Xenopus laevis* embryos as a model system, the authors show that Anillin interacts with Rho forming a feedback loop at the junction: Rho drives Anillin localization and in turn Anillin ensures Rho accumulation. Anillin knockdown in *Xenopus* epithelial tissue leads to severe disruptions to both AJs and TJs as a result of Rho-dependent F-actin perturbation and Myo-II misregulation at cell junctions [207]. This study provides further evidence to suggest that the regulation of Rho activity at the apical junctional complex relies on many of the same proteins that are responsible for regulating Rho activity during cytokinesis, including the centralspindlin complex, MgcRacGAP, Ect2, ARHGEF2 and p190 RhoGAP.

In reality, the cross talk between Rho GTPases and cell junctions is not as straightforward as presented. Although Rho activity can promote junction stability, as previously discussed, high Rho signalling can lead to junction disruption [208–211].

Rho has two major effectors, namely ROCK and Diaphanous-related formin-1 (Dia). Dia, being a formin, promotes linear F-actin elongation and plays a role in stress fibre and filopodia formation. Its activation at the AJ is important [212] and Dia has been shown to be able to sense and generate mechanical forces on actin filaments [213]. The Rho effector kinases ROCKI and ROCKII promote both actomyosin contractility and actin polymerization. Decreased Dia or increased ROCK activation can disrupt AJ integrity [209] indicating that a tight balance between Dia and ROCK activities is required at the junction, with hyperactivity of ROCK leading to junction disruption. Junction stability is not only likely to be regulated by the level of Rho activation but also by the Rho isoform that is activated. For example, RhoC has a greater affinity for ROCK than RhoA [209]. RhoB, on the other hand has been shown to play a role in F-actin distribution: actin filaments in RhoB depleted cells accumulated preferentially in stress fibres rather than at the cell cortex; a phenotype associated with decreased E-cadherin localization at the AJ [214]. These studies highlight the fact that not only is Rho GTPase activity likely to be required to be fine-tuned both spatially and temporally, but the activity and stability of specific GEFs and GAPs is also likely to be critical for correct junction assembly, maturation, homeostasis and for specific morphogenetic movements. The large number of GEFs and GAPs within the cell is conducive to

extremely complex regulatory mechanisms which we are only just beginning to untangle.

# 7. Molecular mechanisms of protein turnover at cell–cell junctions

Cell junctions are highly dynamic structures that continually form and disassemble; an attribute that is essential to preserve epithelial integrity. This inherent junctional plasticity is required to accommodate changes in internal and external mechanical forces over time, and to maintain tissue homeostasis in stable epithelia. During complex morphogenetic events, such as cell intercalation or gastrulation, the necessity for this plasticity is even more evident [6]. Therefore, cell–cell junctions are required to be both strong and plastic and this is achieved through the active turnover of junction components. Several proteins localized at the apical junctional complex are internalized through clathrin, caveolin or dynamin mediated endocytosis and their concentration at the junction reflects the finely tuned mechanism for sorting vesicles for recycling or degradation. E-cadherin turnover at the AJ (figure 3) has been most heavily researched, but other junctional proteins, such as α-catenin, β-catenin, ZO1, Fasciclin III, Crumbs, Pals1, Par3 and claudins undergo similar dynamic recycling [7–12]. The processes involved in E-cadherin and TJ protein turnover have been recently reviewed [13,14] as has the specific role of the Cdc42-Par6-aPKC polarity module in regulating E-cadherin endocytosis [6,14,63]; therefore, this section will simply highlight the key interactions and events, focusing on E-cadherin turnover unless specified otherwise.

Clathrin-mediated endocytosis can be dissected into five stages: nucleation, cargo selection, clathrin coat assembly, vesicle scission and clathrin uncoating [215]. Nucleation (figure 3b) involves the formation of a membrane invagination, which then recruits the adapter protein AP2 [216], which can then bind other cargo-specific adaptor proteins, thereby mediating cargo selection (figure 3c). Once detached from the plasma membrane, following vesicle scission (figure 3d), Hrs/ESCRT mediate sorting processes which target the protein either for recycling or for degradation [217] (figure 3e); the vesicles make use of the microtubule and actin networks to travel back to the junction or towards lysosomes depending on the fate decided for the cargo. Fate is determined pre-internalization by modifications to protein structure (figure 3a). For instance, cleavage of the β-catenin binding domain on the cytoplasmic tail of VE-cadherin by calpain labels VE-cadherin for degradation and increases clathrin-dependent endocytosis rate [218]. In addition, E-cadherin and β-catenin phosphorylation by tyrosine kinases (e.g. Src) enables cadherin–catenin ubiquitylation by E3 ubiquitin-ligases and activates Rab5 and Rab7 GTPases, hence inducing endocytosis and degradation of the labelled proteins [217,219,220].

Extensive data suggest that p120ctn is an important regulator of cadherin endocytosis. In the absence of p120ctn, cadherins are rapidly internalized from the cell surface and degraded in the lysosome [221,222]. The JMD of classical cadherins, the region that binds p120ctn, incorporates a highly conserved endocytic motif. p120ctn binding physically occupies this motif, thereby inhibiting cadherin endocytosis. Therefore, p120ctn binding and cadherin endocytosis are

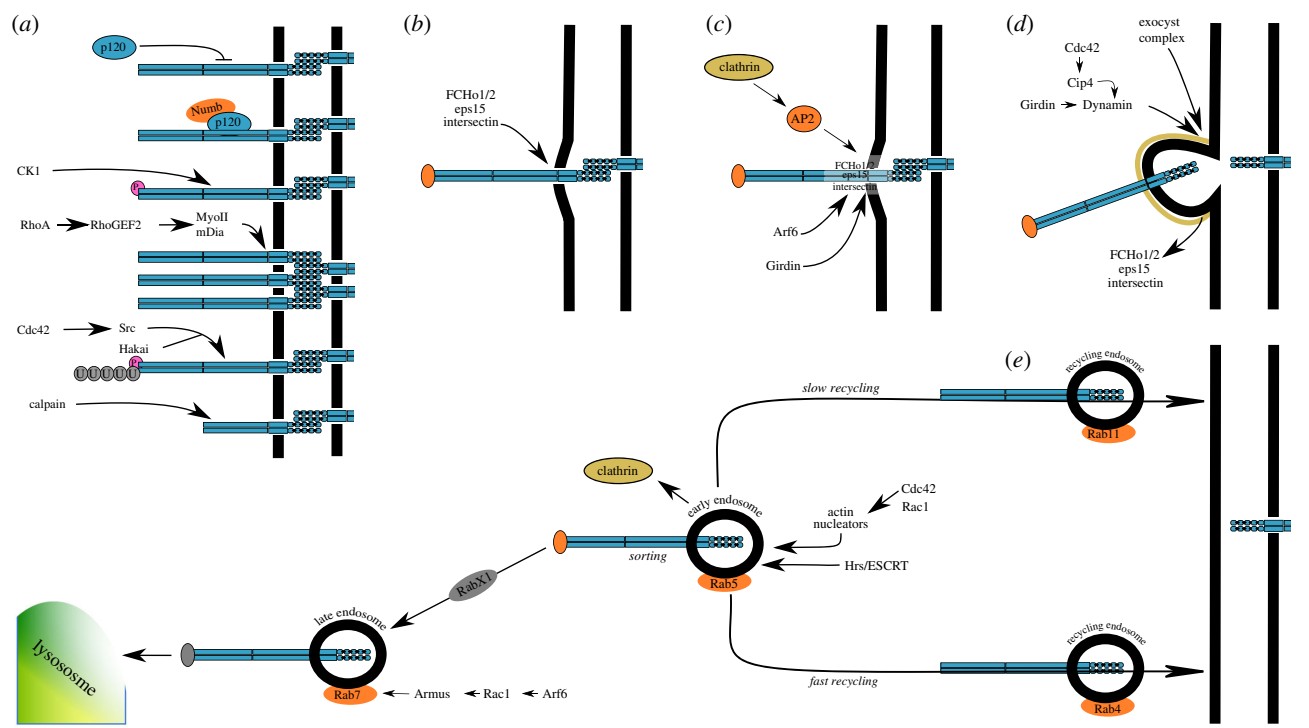

**Figure 3.** Clathrin-mediated cadherin turnover at adherens junctions. (*a*) Protein labelling for endocytosis. Several mechanisms are in place for labelling cadherins for endocytosis. When bound to the juxtamembrane domain, p120-catenin stabilizes cadherin at AJs by masking the binding site for the endocytic machinery, therefore p120 detachment from AJs renders E-cadherin susceptible for internalization. However, Numb interaction with p120 at cell adhesion sites leads to E-cadherin labelling for endocytosis. CK1 mediated E-cadherin phosphorylation impedes E-cadherin–β-catenin interactions and therefore promotes endocytosis. RhoGEF2-activated Myo-II and mDia can induce lateral clustering of E-cadherin which leads to E-cadherin internalization. The 'endocytosis' label added to cadherin can also pre-determine fate after internalization. For instance, Cdc42-activated Src kinase can phosphorylate E-cadherin, thus promoting its ubiquitylation by the E3 ubiquitin-ligase Hakai; calpain can cleave the C-terminal domain of VE-cadherin cytoplasmic tail, hence labelling the protein for degradation. (*b*) Nucleation. F-BAR containing proteins can detect and bind slight curvatures in the cell membrane, which will trigger the assembly of a nucleation module. (*c*) Cargo selection and clathrin coat assembly. The nucleation module will then recruit the AP2 adaptor protein (which can simultaneously bind the cargo protein and the cell membrane) to the nucleation site, where it will further recruit clathrin and cargo-specific adaptor proteins (e.g. girdin) to the junctions. (*d*) Vesicle scission. The endocytic machinery forms a pit around the tagged protein, as F-BAR proteins are excluded from the endocytic site. Further on, girdin-activated dynamin in collaboration with Cip4 and sec10 (a component of the exocyst complex) mediate the scission of the pit from the membrane thereby forming cytoplasmic vesicles known as early endosomes. (*e*) Clathrin uncoating and cargo fate. Once internalized, clathrin disassembles from the endocytic vesicles and cadherins undergo sorting processes mediated by Hrs/ESCRT. Early endosomes associate with Rab5 and recruit Rac1, Cdc42 and actin nucleators on their surface, which allows them to modulate cytoskeleton dynamics in order to facilitate transport. After sorting, cargo proteins can face two different fates: cadherins can either be moved to recycling endosomes associated with Rab11/ Rab4 for slow/fast recycling back to the membrane; or they can be transported in a RabX1-dependent way to late endosomes associated with Rab7. Arf6-activated Rac1 can modulate Rab7 activity by interacting with Armus, a protein that facilitates late endosome cargo transport and degradation in lysosomes.

mutually exclusive events, meaning that p120ctn can trigger cadherin endocytosis simply by dissociating from the cadherin JMD.

Mutation of the endocytic motif was shown to uncouple VE-cadherin from p120ctn while simultaneously preventing the endocytosis of cadherin at the cell surface [223]. It was additionally shown in mammalian cells that the cell fate determinant Numb interacts with the E-cadherin/p120ctn complex and promotes E-cadherin endocytosis. aPKC, a member of the PAR polarity module, was shown to phosphorylate Numb and inhibit its association with p120ctn and AP2, thereby inhibiting E-cadherin endocytosis [224]. This provides a further example of the strong link between polarity and adhesion, whereby polarity proteins can regulate E-cadherin trafficking.

In *Drosophila*, it was shown that p120ctn can either promote or inhibit E-cadherin endocytosis, depending on how much p120ctn is present at the plasma membrane [225]. Here, it was shown that p102ctn can regulate two GTPases, RhoA and Arf1, which work antagonistically in this respect.

RhoA inhibits endocytosis by increasing cortical tension, thereby preventing the membrane bending that is required for vesicle formation and endocytosis [226,227]. Conversely, Arf1 can promote endocytosis by promoting the cytoskeleton remodelling required for vesicle formation [228].

As mentioned at the start of this section, junctional plasticity is required to accommodate changes in mechanical forces over time. It would therefore be expected that turnover at the junction should be mechanosensitive, in order to adapt to these changing forces. De Beco and colleagues [229] directly addressed this hypothesis. Using two-colour two-photon FRAP on MDCK cells, they showed that although turnover rates vary considerably in space and over time, they match remarkably well across individual cell–cell junctions between neighbouring cells. Additionally, using both mechanical and biochemical perturbations, they observed faster turnover rates with increased tension. Mechanical stress was induced perpendicular to cell–cell junctions by pulling on the apical surface with a micropipette, and increased E-cadherin turnover was seen as a direct response; while asymmetric

**Table 1.** Key remodelling processes that may occur in epithelial tissues.

| remodelling processes | examples of physiological processes | key molecular events | key proteins involved | key references |
|---|---|---|---|---|
| new junction formation | cell division | non-autonomous actomyosin tension actin polymerization | Septin; Anilin; Arp2/3 | [234–236] |
| removal of cell–cell junctions | cell death cell extrusion | cell intercalation (T1 swaps) actomyosin supracellular ring contraction | Rho1; Myo-II | [234,237,238] |
| cell intercalation: tetrad formation | *Drosophila* germband extension (initial steps) | dorsoventral/ anteroposterior asymmetry in cell adhesion and contractility polarized distribution of proteins at the AJ contractility at junctional actomyosin network | Myo-II; F-actin (short junctions) E-cad; Arm; Baz (long junctions) Rok | [6,76,239,240] |
| cell intercalation: rosette formation | *Drosophila* germband extension (final steps) vertebrate neural tube or primitive streak formation | asymmetry in cell adhesion and contractility polarized distribution of proteins at the AJ junctional actomyosin network contraction | Shroom/ROCK; Myo-II | [6,76,241,242] |
| apical constriction | *Drosophila* gastrulation | contractility of the randomly oriented medial actomyosin network (pulses) | Rho1/Rok1 pathway; Myo-II; Canoe | [6,76,243] |
| | *Drosophila* dorsal closure | repeated assembly and disassembly of the randomly oriented actomyosin network | Rho1/Rok1 pathway; Myo-II; Canoe; Par complex | [6,244] |
| basal oscillating constriction | *Drosophila* egg chamber elongation | temporary oscillating actomyosin constrictions of the basal actomyosin fibres, organized parallel to the dorsoventral axis | Rho cadherin-mediated adhesion cell–ECM interactions | [6,245] |
| collective cell migration | gastrulation | cadherin internalization | TGFβ pathway; Rab5; dynamin (*Xenopus*) Wnt pathway; Rab5; dynamin (Zebrafish) snail signalling; p38MAPK; p38IP (mouse) | [76] |
| | *Drosophila* wing development | cadherin internalization | PCP signalling; Rab11 | [6,76,246] |
| | cell intercalation in *Drosophila* trachea | cadherin internalization | Rab5 | [6,247] |

mechanical stress was induced by microinjecting constitutively active RhoA into individual cells, which led to increased E-cadherin turnover only in neighbouring cells [229]. In context with substantial literature on junction stabilization as a response to mechanical stress, the authors postulated that low levels of mechanical stress applied at junctions might help stabilize E-cadherin, while a more intense exposure could promote E-cadherin turnover [229]. Very recently, a study performed in the *Drosophila* pupal wing shows that the p120ctn interaction with E-cadherin is lost upon increased mechanical stress with a consequent increase in E-cadherin internalization. The mechanisms behind the E-cadherin–p120ctn dissociation are yet to be determined, but might involve post-translational modifications to p120ctn or, more likely, a mechanically induced conformational change in E-cadherin ternary structure. Regardless, increased E-cadherin turnover confers cells increased viscoelasticity [230]. Importantly, both studies

highlight the benefit of tightly regulated endocytosis for tissue morphogenesis and homeostasis.

# 8. Concluding remarks

Cell–cell adhesion is a fundamental requirement to form the sheets of cells that make up epithelial and endothelial tissues. The common theme emerging from the studies highlighted in this review is the enormous molecular complexity involved in establishing, regulating and maintaining these cell–cell junctions. The transmembrane proteins that physically connect neighbouring cells is of course key to the formation of the junction, but just as important are the regulatory processes that ensure that the correct transmembrane proteins are delivered to the right location, at the right time, at the right concentrations, and which are subsequently maintained at the plasma membrane for the correct length of time. This is where the interconnectivity of the many AJ and TJ cytoplasmic plaque components, cytoskeletal regulators, polarity proteins, Rho GTPases, together with their GEFs and GAPs, as well as the regulators of endocytosis and recycling play their essential role. The concentrations and activities of these many molecules, which are highly dynamic and which impinge on one another to a great degree, are not only essential to maintain cell–cell adhesion, but also cell polarity and cell morphology. It is quite astonishing to consider that this complex interplay, this highly sophisticated molecular dance between so many performers, is required simply to maintain intercellular junctions, even in stable epithelia. However, cells possess the ability to carry out numerous cell shape changes and this is where this molecular complexity becomes essential. The coordinated cell movements that are important for organism growth and development, as well as for tissue repair, require not only complex cell shape changes but also changes to cell–cell junction length, processes controlled through the regulation of cell–cell adhesion and contractility [231]. In this context, junction dynamics are a key factor in enabling these morphogenetic processes [6]. Epithelial cells possess the molecular machinery to carry out multiple morphogenetic processes, ready to be used should they so be required. Depending on the tissue and the stage of development, different cell shape changes could be achieved by employing different adhesion and/or constriction regulators, for example, forcing apical constriction during gastrulation [232], or focusing constriction forces to a single junction, required during cell intercalation [233]. The key remodelling interactions that can occur in an epithelial tissue are summarized in table 1. We direct the reader towards some excellent reviews that discuss this matter in greater detail [6,76,234,248].

Importantly, one consequence of failing to correctly regulate cell–cell junctions is cancer. Since cell–cell adhesion, cell polarity and cell shape are so tightly interlinked, a loss of adhesion can lead to tumour progression towards malignancy. In order for tumours to become malignant they must become invasive, and this requires epithelial cells to take on mesenchymal or amoeboid characteristics [249]. However, in order to form a secondary tumour at a secondary site, cells must undergo the opposite process and regain their epithelial characteristics, in order to insert into a new tissue at the new site. Each of these processes relies on the ability of cells to manipulate their adhesive state. It is therefore no surprise to find that the same collection of proteins that regulate cell–cell junctions (regulators of adhesion, polarity, the cytoskeleton, trafficking) are also implicated in cancer and metastasis [63,250–253].

Data accessibility. This article has no additional data.
Authors' contributions. A.D.R. and M.G. wrote the paper.
Competing interests. We declare we have no competing interests.
Funding. Our laboratory received funding from Cancer Research UK (grant nos C36430, A12891) and NC3Rs (grant no. NC/S001417/1). A.D.R. was supported by a Nottingham Vice-Chancellor's Scholarship for Research Excellence Award.

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
