## [Reviewer comments · Open Biology]

Review History

RSOB-19-0278.R0 (Original submission)

Review form: Reviewer 1

Recommendation

Accept with minor revision (please list in comments)

Do you have any ethical concerns with this paper?

No

Comments to the Author

This review focuses on regulation and composition of cell-cell adhesion, specifically adherens and tight junctions. It covers a broad range of topics including interplay between two junction types, regulation by Rho GTPases and endocytosis at a detailed, state-of-the-art level. Overall, it is well written and enjoyable to read.

1. On page 5, the relevance of the paragraph about TJ ultrastructure on ultrathin sections is unclear, it does not add any valuable information and breaks the flow, so I suggest removing it, or if keeping, then placing it in a different location earlier in the text. An additional image could be helpful to visualize the strands.
2. An illustration of a tTJ could be also helpful.
3. Similarly, the reasons for inclusion of ultrastructure of AJs on page 9 are unclear.

4. On page 10, the statement that beta-catenin seems to be dispensable for AJ adhesive function is not fully justified as it is still needed in normal circumstances to connect E-cadherin to alpha-catenin.
5. It is unclear why among less studied members of the cadherin-catenin junctional complex, vinculin is included, while other proteins such as EPLIN.
6. The first sentence on page 10 and the last sentence in the second paragraph on page 11 are virtually identical – I suggest removing one of them.
7. Page 17: should be EPLIN instead of EPLN.
8. “de novo” on page 14, “in vitro” on page 17, and “in vivo” on page 18 should be italicised.
9. References seem to be missing in the second paragraph on page 22.
10. The order of figures is rather awkward with figure 4 being cited before figure 3.

Review form: Reviewer 2

Recommendation

Major revision is needed (please make suggestions in comments)

Do you have any ethical concerns with this paper?

No

Comments to the Author

The review by Rusu and Georgiou titled “ The multifarious regulation of the apical junctional complex” attempts to give a comprehensive summary of the regulation of junctional proteins by cytoskeletal dynamics and trafficking. This review provides a list of junctional regulators, however these interactions are not necessarily summarised in the context of physiological processes. Remodelling mechanisms could be tissue specific and all interactions may not hold as a generic principle and hence I recommend some edits or additions to give importance to the physiological process that is referred to in each context of remodelling before publication. I also recommend focussing the review to a few processes so that it provides a specific compilation in only those processes. Some points for revision are summarized below:

1. Clarify the goal of the review, does it focus on recent advances, are there any new advances which motivate this review in the current time.
2. What are the physiological processes that require junction formation and remodelling that the review has derived the literature from? Can remodelling processes be picked based on the process that it is involved. Does every remodelling process require all mechanisms and molecules to be a part of it.
3. The review misses the opportunity to summarise the key remodelling interactions in a tabular format along with the figures that depicted.
4. I recommend a thorough check of spellings and typos in the writing and figures: Figure 2C, please check the spelling of actomyosin ring.
5. Figure 1a and b where the vertebrate and invertebrate junctions are compared deserves the opportunity of distinguishing molecular differences within the cell also.
6. What would be the difference between microfilaments and actin filaments in the Figure 1 be?, what is the need to show them separately, just actin filaments will suffice?
7. Figure 1: Shroom2 is the protein that regulates junction and not Sroom2?
8. What is the difference between junction expansion and junction maturation? Is this trying to distinguish 2 steps where binding between extracellular domains occurs and binding is consolidated or remodelling is prevented?
9. Figure 2 and 4 could be combined to give a more digested view of the literature for the role of actin regulators and small GTPases.

Decision letter (RSOB-19-0278.R0)

20-Dec-2019

Dear Dr Georgiou,

We are pleased to inform you that your manuscript RSOB-19-0278 entitled "The multifarious regulation of the apical junctional complex" has been accepted by the Editor for publication in Open Biology. The reviewer(s) have recommended publication, but also suggest some revisions to your manuscript. Therefore, we invite you to respond to the reviewer(s)' comments and revise your manuscript.

Please submit the revised version of your manuscript within 21 days. If you do not think you will be able to meet this date please let us know immediately and we can extend this deadline for you.

- 1) A text file of the manuscript (doc, txt, rtf or tex), including the references, tables (including captions) and figure captions. Please remove any tracked changes from the text before submission. PDF files are not an accepted format for the "Main Document".
- 2) A separate electronic file of each figure (tiff, EPS or print-quality PDF preferred). The format should be produced directly from original creation package, or original software format. Please note that PowerPoint files are not accepted.
- 3) Electronic supplementary material: this should be contained in a separate file from the main text and meet our ESM criteria (see <http://royalsocietypublishing.org/instructions-authors#question5>). All supplementary materials accompanying an accepted article will be treated as in their final form. They will be published alongside the paper on the journal website and posted on the online figshare repository. Files on figshare will be made available approximately one week before the accompanying article so that the supplementary material can be attributed a unique DOI.

Online supplementary material will also carry the title and description provided during submission, so please ensure these are accurate and informative. Note that the Royal Society will not edit or typeset supplementary material and it will be hosted as provided. Please ensure that the supplementary material includes the paper details (authors, title, journal name, article DOI). Your article DOI will be 10.1098/rsob.2016[*last 4 digits of e.g. 10.1098/rsob.20160049*].

4) A media summary: a short non-technical summary (up to 100 words) of the key findings/importance of your manuscript. Please try to write in simple English, avoid jargon, explain the importance of the topic, outline the main implications and describe why this topic is newsworthy.

Images

Data-Sharing

It is a condition of publication that data supporting your paper are made available. Data should be made available either in the electronic supplementary material or through an appropriate repository. Details of how to access data should be included in your paper. Please see <http://royalsocietypublishing.org/site/authors/policy.xhtml#question6> for more details.

Data accessibility section

Sincerely,

The Open Biology Team

<mailto:openbiology@royalsociety.org>

Reviewer(s)' Comments to Author:

Referee: 1

Comments to the Author(s)

This review focuses on regulation and composition of cell-cell adhesion, specifically adherens and tight junctions. It covers a broad range of topics including interplay between two junction types, regulation by Rho GTPases and endocytosis at a detailed, state-of-the-art level. Overall, it is well written and enjoyable to read.

1. On page 5, the relevance of the paragraph about TJ ultrastructure on ultrathin sections is unclear, it does not add any valuable information and breaks the flow, so I suggest removing it, or if keeping, then placing it in a different location earlier in the text. An additional image could be helpful to visualize the strands.
2. An illustration of a tTJ could be also helpful.
3. Similarly, the reasons for inclusion of ultrastructure of AJs on page 9 are unclear.
4. On page 10, the statement that beta-catenin seems to be dispensable for AJ adhesive function is not fully justified as it is still needed in normal circumstances to connect E-cadherin to alpha-catenin.
5. It is unclear why among less studied members of the cadherin-catenin junctional complex, vinculin is included, while other proteins such as EPLIN.
6. The first sentence on page 10 and the last sentence in the second paragraph on page 11 are virtually identical – I suggest removing one of them.
7. Page 17: should be EPLIN instead of EPLN.

8. "de novo" on page 14, "in vitro" on page 17, and "in vivo" on page 18 should be italicised.
9. References seem to be missing in the second paragraph on page 22.
10. The order of figures is rather awkward with figure 4 being cited before figure 3.

Referee: 2

Comments to the Author(s)

The review by Rusu and Georgiou titled "The multifarious regulation of the apical junctional complex" attempts to give a comprehensive summary of the regulation of junctional proteins by cytoskeletal dynamics and trafficking. This review provides a list of junctional regulators, however these interactions are not necessarily summarised in the context of physiological processes. Remodelling mechanisms could be tissue specific and all interactions may not hold as a generic principle and hence I recommend some edits or additions to give importance to the physiological process that is referred to in each context of remodelling before publication. I also recommend focussing the review to a few processes so that it provides a specific compilation in only those processes. Some points for revision are summarized below:

1. Clarify the goal of the review, does it focus on recent advances, are there any new advances which motivate this review in the current time.
2. What are the physiological processes that require junction formation and remodelling that the review has derived the literature from? Can remodelling processes be picked based on the process that it is involved. Does every remodelling process require all mechanisms and molecules to be a part of it.
3. The review misses the opportunity to summarise the key remodelling interactions in a tabular format along with the figures that depicted.
4. I recommend a thorough check of spellings and typos in the writing and figures: Figure 2C, please check the spelling of actomyosin ring.
5. Figure 1a and b where the vertebrate and invertebrate junctions are compared deserves the opportunity of distinguishing molecular differences within the cell also.
6. What would be the difference between microfilaments and actin filaments in the Figure 1 be?, what is the need to show them separately, just actin filaments will suffice?
7. Figure 1: Shroom2 is the protein that regulates junction and not Sroom2?
8. What is the difference between junction expansion and junction maturation? Is this trying to distinguish 2 steps where binding between extracellular domains occurs and binding is consolidated or remodelling is prevented?
9. Figure 2 and 4 could be combined to give a more digested view of the literature for the role of actin regulators and small GTPases.

Author's Response to Decision Letter for (RSOB-19-0278.R0)

See Appendix A.

Decision letter (RSOB-19-0278.R1)

27-Jan-2020

Dear Dr Georgiou,

We are pleased to inform you that your manuscript entitled "The multifarious regulation of the apical junctional complex" has been accepted by the Editor for publication in Open Biology.

Sincerely,

The Open Biology Team
mailto: openbiology@royalsociety.org

Appendix A

Response to reviewers.

First, we would like to thank the reviewers for reviewing our paper. Having revised the manuscript to take the reviewers comments into account, we believe we have a stronger paper and we thank the reviewers for their input. Our point-by-point response is detailed below.

Reviewer #1:

This review focuses on regulation and composition of cell-cell adhesion, specifically adherens and tight junctions. It covers a broad range of topics including interplay between two junction types, regulation by Rho GTPases and endocytosis at a detailed, state-of-the art level. Overall, it is well written and enjoyable to read.

1. On page 5, the relevance of the paragraph about TJ ultrastructure on ultrathin sections is unclear, it does not add any valuable information and breaks the flow, so I suggest removing it, or if keeping, then placing it in a different location earlier in the text. An additional image could be helpful to visualize the strands.

This paragraph has now been moved.

2. An illustration of a tTJ could be also helpful.

We have added a schematic diagram of a tTJ to Figure 1 (Figure 1d)

3. Similarly, the reasons for inclusion of ultrastructure of AJs on page 9 are unclear.

This section has been removed.

4. On page 10, the statement that beta-catenin seems to be dispensable for AJ adhesive function is not fully justified as it is still needed in normal circumstances to connect E-cadherin to alpha-catenin.

We agree with the reviewer's point and have now rephrased this sentence.

5. It is unclear why among less studied members of the cadherin-catenin junctional complex, vinculin is included, while other proteins such as EPLIN.

EPLIN's role at the junction has now been added to this paragraph.

6. The first sentence on page 10 and the last sentence in the second paragraph on page 11 are virtually identical - I suggest removing one of them.

The latter sentence has been removed.

7. Page 17: should be EPLIN instead of EPLN.

Corrected.

8. "de novo" on page 14, "in vitro" on page 17, and "in vivo" on page 18 should be italicised.

Corrected.

9. References seem to be missing in the second paragraph on page 22.

References added.

10. The order of figures is rather awkward with figure 4 being cited before figure 3.

Figure 4 has now been moved to Figure 2 (it is now Figure 2d).

Reviewer #2:

1. Clarify the goal of the review, does it focus on recent advances, are there any new advances which motivate this review in the current time.

The goal of the review has now been clarified in the Introduction.

2. What are the physiological processes that require junction formation and remodelling that the review has derived the literature from? Can remodelling processes be picked based on the process that it is involved. Does every remodelling process require all mechanisms and molecules to be a part of it.

The point that we tried to make in the review article, and which we have now expanded upon in the Concluding Remarks, is that epithelial cells possess the molecular machinery to carry out multiple morphogenetic processes, ready to be utilised if and when required.

3. The review misses the opportunity to summarise the key remodelling interactions in a tabular format along with the figures that depicted.

We have now added a table (Table 1) summarising key remodelling events and the key proteins involved.

4. I recommend a thorough check of spellings and typos in the writing and figures: Figure 2C, please check the spelling of actomyosin ring.

This has been done.

5. Figure 1a and b where the vertebrate and invertebrate junctions are compared deserves the opportunity of distinguishing molecular differences within the cell also.

Molecular detail has now been added to these panels.

6. What would be the difference between microfilaments and actin filaments in the Figure 1 be?, what is the need to show them separately, just actin filaments will suffice?

Figure 1 has been changed accordingly.

7. Figure 1: Shroom2 is the protein that regulates junction and not Sroom2?

Corrected.

8. What is the difference between junction expansion and junction maturation? Is this trying to distinguish 2 steps

where binding between extracellular domains occurs and binding is consolidated or remodelling is prevented?

Junction expansion results through Rac1 activity, which leads to contact expansion and the promotion of further Ecad interactions between neighbouring cells. Junction maturation relies on the assembly of multiple scaffold and signalling proteins at the junction. This is linked to the establishment of apicobasal polarity, the formation of an actomyosin ring at the ZA, and a switch from Rac1 to RhoA signalling. This point has now been clarified in the manuscript.

9. Figure 2 and 4 could be combined to give a more digested view of the literature for the role of actin regulators and small GTPases.

Figure 4 has now been moved to Figure 2 (it is now Figure 2d).